# Excitation energy mediated cross-relaxation for tunable upconversion luminescence from a single lanthanide ion

Xiao Fu[1,7], Shuai Fu[2,7], Qi Lu[1,7], Jing Zhang[2,7], Pingping Wan[3], Jinliang Liu[2], Yong Zhang [4] ✉, Chia-Hung Chen [5], Wei Li[1], Huadong Wang[6] & Qingsong Mei [1,6] ✉

Precise control of energy migration between sensitizer ions and activator ions in lanthanide-doped upconversion nanoparticles (UCNPs) nowadays has been extensively investigated to achieve efficient photon upconversion. However, these UCNPs generally emit blue, green or red light only under fixed excitation conditions. In this work, regulation of the photon transition process between different energy levels of a single activator ion to obtain tunable upconversion fluorescence under different excitation conditions is achieved by introducing a modulator ion. The cross-relaxation process between modulator ion and activator ion can be controlled to generate tunable luminescence from the same lanthanide activator ion under excitation at different wavelengths or with different laser power density and pulse frequency. This strategy has been tested and proven effective in two different nanocrystal systems and its usefulness has been demonstrated for high-level optical encryption.

Tuning the luminescence color of a simple material in a broad spectral region upon external stimuli is important because of its great demand in many revolutionary applications including complex information storage, biological imaging, and photoactivations. For instance, compared with traditional encryption strategies such as digital watermark, laser holography, structural colored patterns and so on[1–5], luminescence-based encryption has recently attracted a lot of interest because of its short response time, easy operation, and low cost[6–8]. As of now, different luminescent materials including organic dyes, semiconductor quantum dots, and perovskite nanocrystals, have been explored in encryption applications[9–12]. Lanthanide-doped upconversion nanoparticles (UCNP) with multi-color emissions in ultraviolet, visible, and near-infrared spectral regions have also contributed to this, and they are usually excited under fixed conditions[13–15]. However, these conventional fluorescent materials are simply used as a covert ink to print encrypted

codes[16,17], which can be easily decoded or counterfeited, so there is a great demand for developing tunable and dynamically changeable luminescent materials to improve encryption security.

Precise design and synthesis of UCNPs with a specific optical response in multiple dimensions such as fluorescence color, lifetime, and intensity, pave the way to achieve high-level encryption. For instance, a high concentration of activator ion-doped UCNPs was reported to have a tunable luminescence lifetime for anticounterfeiting[18]. The hidden encrypted patterns were decoded by the expensive time-gated microscopy only, which was not portable and not suitable for practical use. Moreover, orthogonal emissive UCNPs were explored to offer new encryption dimensions and high encoding capacities for anticounterfeiting[19,20]. The multiple emissions of these UCNPs usually do not come from the same lanthanide ion, but from different ions spatially separated by doping in different layers.

[1]Department of Medical Biochemistry and Molecular Biology, School of Medicine, Jinan University, 510632 Guangzhou, Guangdong, China. [2]School of Environmental and Chemical Engineering, Shanghai University, 200444 Shanghai, China. [3]School of Food and Biological Engineering, Hefei University of Technology, 230009 Hefei, Anhui, China. [4]Department of Biomedical Engineering, College of Design and Engineering, National University of Singapore, Singapore 117583, Singapore. [5]Department of Biomedical Engineering, City University of Hong Kong, 83 Tat Chee Avenue, Kowloon, Hong Kong SAR, China. [6]Key Laboratory of State Administration of Traditional Chinese Medicine of the People's Republic of China, School of Medicine, Jinan University, 510632 Guangzhou, Guangdong, China. [7]These authors contributed equally: Xiao Fu, Shuai Fu, Qi Lu, Jing Zhang. ✉e-mail: biezy@nus.edu.sg; qsmei@jnu.edu.cn

Because most of these lanthanide ions ($Er^{3+}$, $Tm^{3+}$, etc.) have multiple emissions, unwanted emissions need to be quenched by adding additives to the particles to obtain orthogonal emissions. Thus, these particles usually have a multi-layered structure and the particle synthesis is tedious[21,22]. Furthermore, these particles are usually excited under fixed conditions.

Although few works have reported tailored emission color of nanocrystals can be obtained via tuning some excitation schemes (e.g., pulse width or pulse frequency)[19,23], it is still a challenge to systematically modulate the excitation conditions of UCNPs to generate different multicolor emissions from a single activator ion. As a nonlinear anti-Stokes process, upconversion luminescence originated from energy migration between different intermediate energy states of lanthanide activator ions[15]. Thus, manipulating the intermediate states through external stimuli should be an effective approach to achieve dynamic luminescence output. It is well-known that cross-relaxation (CR) is an important process in upconversion luminescence, which always occurs between different intermediate energy levels of neighboring emitter ions[24]. Typically, in a CR process, one emitter ion initially in an excited state exchanges photon energy with another ion in the ground state, so both ions are at their intermediate excited states. The energy loss for the first emitter ion is equal to the energy gain for the second ion, thus the total energy is conserved in the CR process. For example, $Tm^{3+}$ ions heavily doped upconversion nanoparticles always undergo multiple CR pathways, $^1G_4 + ^3H_6 \rightarrow ^3F_{2,3} + ^3F_4$, $^1G_4 + ^3H_6 \rightarrow ^3H_5 + ^3H_4$, $^3F_{2,3} + ^3F_4 \rightarrow ^3H_4 + ^3H_5$[25,26]. Another important activator, $Er^{3+}$ ions, undergo the CR pathways, $^4F_{7/2} + ^6I_{11/2} \rightarrow 2\,^4F_{9/2}$, leading to accumulation of the energy at the intermediate excited states and generation of luminescence emission at a specific wavelength[27,28]. The population distribution of each energy state of these lanthanide ions is strongly dependent on the absorbed excitation energy. Therefore, the CR process is expected to be modulated by multiple parameters such as power density, excitation frequency, or wavelength.

In this work, we propose a concept of introducing a modulator ion into the UCNPs and precisely controlling the CR process between the modulator ion and the activator ion, which is different from controlling the energy transfer from the sensitizer ion to the activator ion as previously reported, so multicolor emissions can be generated from a single activator ion under dynamic and tunable excitation conditions. A core−shell−shell nanoparticle is designed with the outer light-absorbing shell and light-emitting core separated by an inert middle layer. As shown in Fig. 1a, the Nd sensitizer ions for harvesting 808 nm light are doped in the outmost layer of the core−shell−shell nanoparticle while the activator and modulator ions are doped in the core. The middle layer is an energy migration layer, $NaYF_4$:Yb, in which the energy migrating Yb ions act as a valve to control the energy transfer from the outer layer to the core when the 808 nm excitation light is harvested by Nd ions (Fig. 1b). On the other side, the 980 nm excitation light can be directly absorbed by the Yb ions in the core and the two shells. Such a discrepancy leads to different pathways of the energy migration between the modulator and activator ions in the core. These nanoparticles generate different color emissions when excited under different conditions.

## Results

### Luminescence modulation of Ho-Ce-based UCNPs

As a proof-of-concept, we first investigated a typical CR process between Ho ions and Ce ions, in which the former acted as an activator and the latter as a modulator. It is well known that the CR processes of $^5F_4$ ($Ho^{3+}$) + $^2F_{5/2}$ ($Ce^{3+}$) → $^5F_5$ ($Ho^{3+}$) + $^2F_{7/2}$ ($Ce^{3+}$) and $^5I_6$ ($Ho^{3+}$) + $^2F_{5/2}$ ($Ce^{3+}$) → $^5I_7$ ($Ho^{3+}$) + $^2F_{7/2}$ ($Ce^{3+}$) are easy to occur when $Ce^{3+}$ was heavily co-doped with Ho ions[29,30]. However, in the case of Nd-mediated energy migration system, after harvesting 808 nm excitation energy, Nd ions first migrate the absorbed energy to Yb ions in the middle layer, and then transfer the photon energy to Ho ion in the core nanocrystal (Fig. 2a). Therefore, the population of excited Ho ions is highly dependent on the energy migration by Yb ions.

The influence of Yb ion concentrations on the CR efficiency in the core layer was detailedly studied. As shown in Fig. 2b, the $NaNdF_4$:Yb layer was directly coated on the CR layer, $NaYF_4$:Yb/Ho/Ce. It is found that these core−shell structured nanoparticles exhibit red-colored emission no matter excited at 980 or 808 nm lasers. Both emission profiles showed that red emission intensities were about 2.9-fold higher than green emission intensities, which should be ascribed to the fact that CR has occurred between Ho and Ce ions at both exciting conditions. This means that excitation energy of 808 nm laser is highly efficient migrated into core nanocrystal, which is just like the energy absorption and transfer processes that happened in core counterpart upon 980 nm excitation. Interestingly, the luminescent properties were absolutely different when inserting an energy migration layer, $NaYF_4$:Yb, between them. As shown in Fig. 2d and Supplementary Fig. 1, the obtained core−shell−shell structured nanoparticles, $NaYF_4$:Yb/Ho/Ce@$NaYF_4$:Yb@$NaNdF_4$:Yb, demonstrated a dumbbell-shaped morphology. X-ray diffraction patterns in Supplementary Fig. 2 indicated that the obtained UCNPs were indexed to hexagonal phase nanocrystals. Elemental mapping results showed that Nd ions were located at both ends of the dumbbell, while Ho/Ce ion pairs were located at the center of the dumbbell (Supplementary Fig. 3). Thus, the excitation energy transfer process of 808 nm laser went a longer pathway, decreasing the populations of $^5F_5$ and $^5I_7$ states. Thereby, the dumbbell-shaped UCNP mainly emitted green-colored light upon 808 nm excitation. On the contrary, Yb ions in the core nanocrystals could directly harvest 980 nm excitation energy, and the energy transfer distance was shorter, resulting in giving out CR-mediated red-

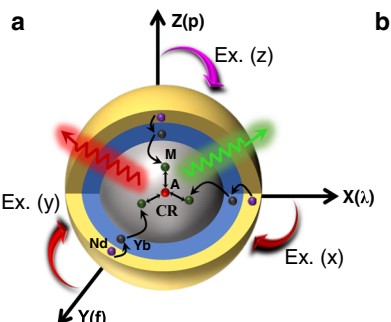
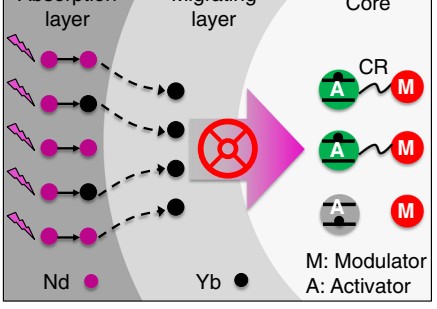
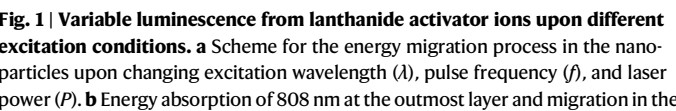

**Fig. 1 | Variable luminescence from lanthanide activator ions upon different excitation conditions. a** Scheme for the energy migration process in the nanoparticles upon changing excitation wavelength ($\lambda$), pulse frequency ($f$), and laser power ($P$). **b** Energy absorption of 808 nm at the outmost layer and migration in the middle layer influence the CR efficiency between activator ions and modulator ions in the core nanocrystal. Migrating layer plays as an energy valve to control energy transition from the outer layer to the core layer.

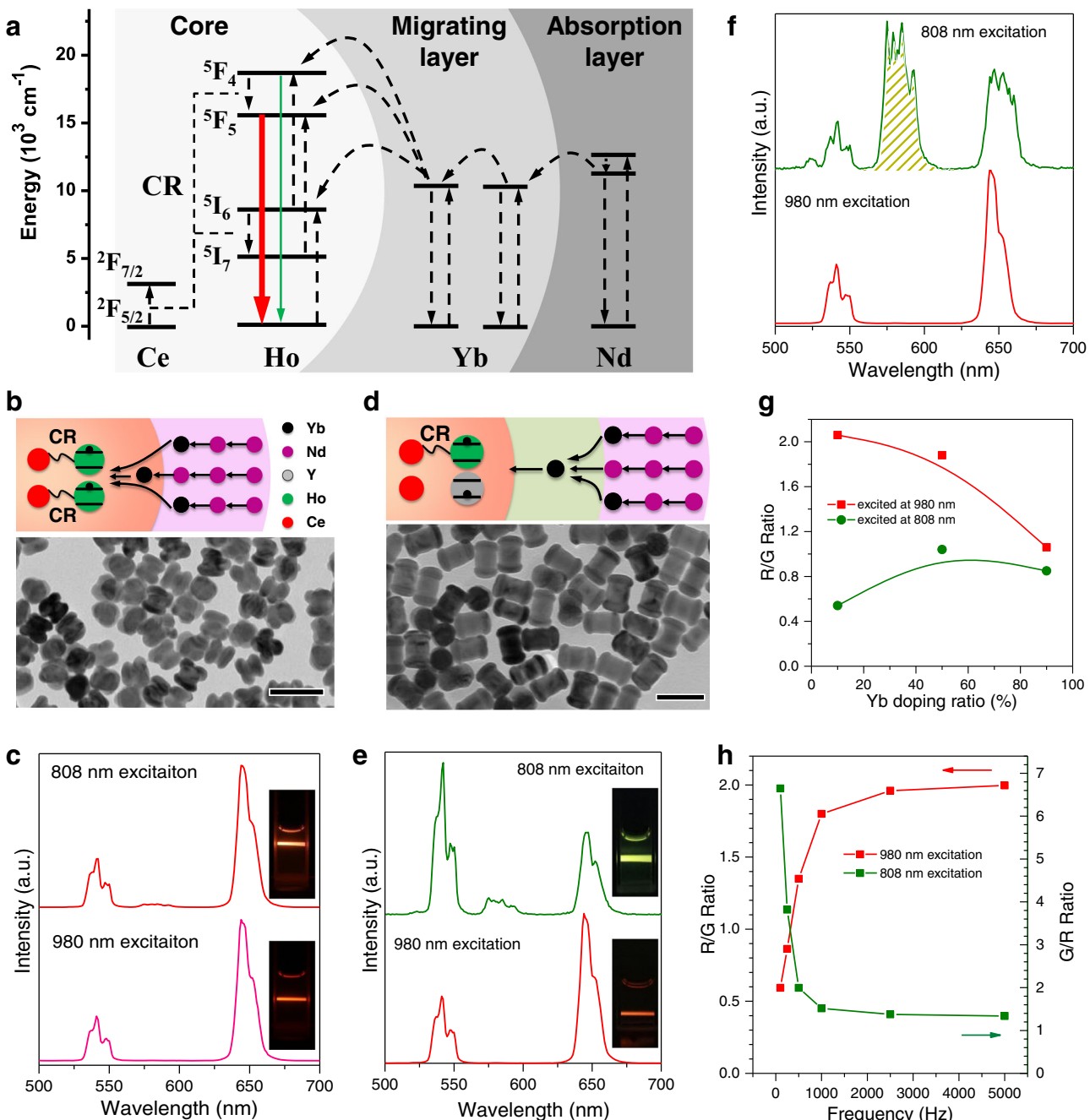

**Fig. 2 | Luminescence modulation of the Ho–Ce system. a** Schematic illustration of the energy migration process in NaYF$_4$:Yb/Ho/Ce@NaYF$_4$:Yb@NaNdF$_4$:Yb nanocrystals upon excitation with 808 nm. **b** Scheme illustration of the energy migration process and TEM image of NaYF$_4$:Yb/Ho/Ce@ NaNdF$_4$:Yb. **c** Luminescent spectra of NaYF$_4$:Yb/Ho/Ce@ NaNdF$_4$:Yb under 980 and 808 nm excitation. Inset images are corresponding luminescent photos. **d** Schematic illustration of migrating layer mediated energy transfer process and TEM image of NaYF$_4$:Yb/Ho/ Ce@NaYF$_4$:Yb@NaNdF$_4$:Yb. **e** Luminescent spectra of the above-mentioned nanoparticles under 980 and 808 nm excitation. Inset images are corresponding luminescent photos. **f** Luminescent spectra of the nanoparticles with no sensitizer ion doped in migrating layer, NaYF$_4$:Yb/Ho/Ce@NaYF$_4$@NaNdF$_4$:Yb, under 980 and 808 nm excitation. **g** Intensity ratios of red emission to green emission of the nanoparticles NaYF$_4$:Yb/Ho/Ce@NaYF$_4$:Yb@NaNdF$_4$:Yb after changing Yb doping concentrations in the middle migrating layer. **h** Intensity ratios of red to green upon 980 nm excitation and ratios of green to red upon 808 nm excitation of the nanoparticles NaYF$_4$:Yb/Ho/Ce@NaYF$_4$:Yb@NaNdF$_4$:Yb after changing excitation frequencies. All scale bars in TEM images are 100 nm. Source data in Fig. 2c, Fig. 2e–h are provided as a source data file.

dominated emission, which was about 2.3-fold higher than green emission intensity (Fig. 2e).

In the cross-relaxation process, the doping amount of activator ions and modulator ions plays a crucial role in CR efficiency[31]. Herein, the CR process between Ho and Ce ions leads to increased populations of the excited levels of $^5F_5$ and $^5F_7$ which can give out red emissions. A wide range of doping concentrations of Ho ions or Ce ions in the core nanocrystals were investigated. As shown in Supplementary Fig. 6-7, to

make the greatest difference in emission colors at 980 and 808 nm excitation, the ideal proportion of Ce ions was established to be 15%. Similarly, the optimal doping ratio of Ho ions in the core was chosen to be 1.5% (Supplementary Figs. 8 and 9).

To further verify the energy valve function of the migrating layer, different amounts of Yb ions in this layer were doped to study the luminescent variations. It was noted that the 808 nm excitation energy was not able to migrate into core nanocrystals when no Yb ion was

doped in the middle layer. Intriguingly, a totally different emission profile was presented with a new peak at 580 nm, which mainly be ascribed to the luminescence from Nd ions (Fig. 2f). However, it remained red color emission upon excitation at 980 nm. Figure 2g and Supplementary Fig. 11 showed that gradually increasing Yb doping concentrations from 10% to 90% would make emission profiles identical when excitation with 980 or 808 nm laser. This also verified that migrating ions in the middle layer played a significant role in modulating CR efficiency in core nanocrystals when harvesting excitation energy at outer layers.

In this CR circumstance, sensitizer, activator, and modulator are three different ions, the excitation photons transit among these ions with a long-distance pathway. In detail, in comparison with red emissions from the $^5F_5$ state of $Ho^{3+}$ ions arising from the CR process, it is known that green emissions from the $^5F_4$ states can be populated much faster through a direct two-step energy transfer process (Fig. 2a). Thus, the nanoparticles' intensity ratios of red to green emissions are able to be modulated through changing the excitation intervals if a pulse excitation laser was used. Shortening excitation time interval (increasing frequency) significantly increases excitation duration, facilitating the CR process between Ho and Ce ions. In this work, after irradiation at a 150 μs-width pulsed laser with different pulse frequencies, the red/green emission intensity ratio of the nanoparticles, $NaYF_4:Yb/Ho/Ce@NaYF_4:Yb@NaNdF_4:Yb$, was precisely tuned. The intensity enhancement of red emission was found to be much greater than that of green emission upon increasing the frequency from 100 to 5000 Hz (Supplementary Fig. 13).

It is worth noting that, after increasing pulse frequency to 1000 Hz, the depopulated photons at the red emissive energy state can be quickly promoted again by the next pulse excitation light, which is similar to the previously reported non-steady upconversion states[19,23]. On the contrary, depopulated photons at the green emission state almost annihilate when the next pulse is excited because of its lower lifetime (Supplementary Fig. 14). However, the lifetime at 650 nm was about 0.38 ms, and that at 540 nm was 0.21 ms, and those remained unchanged after pulse width changed from 35 to 350 μs with the frequency of 100 Hz (Supplementary Fig. 15). Therefore, by adjusting the pulse frequency, dynamic control of the emission intensity ratios both at 980 and 808 nm excitation was successfully achieved (Fig. 2h), meanwhile the nanoparticles could produce totally different emission colors. For instance, irradiation with a relatively high frequency (5000 Hz) yielded a visible red emission upon excitation with 980 nm. In contrast, the nanoparticles emitted a dominant greenish yellow color when excited with 808 nm laser with the same frequency (Supplementary Fig. 16). Moreover, power dependence investigations indicated that both red and green emission of Ho ions were governed by two-photon excitation processes, and the nanoparticles did not exhibit noticeable changes in emission color (Supplementary Fig. 5).

### Luminescence modulation of Yb-Er-based UCNPs

When sensitizer ions simultaneously play a role as the modulator to tune the emission of activator ions through the CR process, it is expected to demonstrate a few unprecedented phenomena after changing excitation conditions. For instance, the heavily doped Yb ions will undergo a CR pathway with Er ions, following the process $^4G_{11/2}$ ($Er^{3+}$) + $^2F_{7/2}$ ($Yb^{3+}$) → $^4F_{9/2}$ ($Er^{3+}$) + $^2F_{5/2}$ ($Yb^{3+}$)[32,33]. This CR process increases the populations of excitons at $^4F_{9/2}$ states of $Er^{3+}$ ions, increasing red emission intensity. On the basis of this, a core–shell–shell structured nanoparticle, $NaYbF_4:Er@NaYF_4:Yb@NaNdF_4:Yb$, was designed to study the luminescence variations towards the Yb–Er CR system (Fig. 3a and b). The obtained nanoparticles also demonstrated a dumbbell-shaped structure (Fig. 3c). Elemental mapping images in Supplementary Fig. 17 showed that Nd ions grew longitudinally at the ends of the

dumbbell, which should be ascribed to the lattice mismatch between $NaYF_4$ and $NaNdF_4$ lattice.

Luminescence investigations in Fig. 3d showed that a low doping Yb ratio (10%) in the middle layer induced a dominant green emission upon excitation with 808 nm laser. Increasing Yb concentration to 90% markedly raised red light intensity at the same excitation conditions. However, nanoparticles exhibited a typical red-light emission profile under 980 nm excitation no matter 10% or 90% of Yb ions doped in the migrating shell (Supplementary Fig. 18). To further elaborate on the influence of Yb ions in the middle layer towards luminescent efficiencies, $Gd^{3+}$ ions were substituted to dope in the middle layer, obtaining the UCNPs, $NaYbF_4:Er@NaYF_4:Gd(10\%)@NaNdF_4:Yb$. Since $Yb^{3+}$ ions in the middle layer play a pivotal role in the migration of 808 nm excitation photons, which are absorbed by $Nd^{3+}$ ions in the outmost layer, to the activators $Er^{3+}$ in the inner layer, this substitution makes luminescent intensities dramatically decrease upon 808 nm excitation. It is found that intensities of green emission at 540 nm increase about 24 folds when doping with the same amounts of $Yb^{3+}$ ions compared with a dopant of $Gd^{3+}$ ions (Supplementary Fig. 19).

More interestingly, this CR process between Yb and Er ion was highly dependent on excitation power. As exhibited in Fig. 3e, in the case of $NaYbF_4:Er@NaYF_4:Yb(10\%)@NaNdF_4:Yb$, a 20 cm long quartz capillary filled with cyclohexane solution of this nanoparticle demonstrated a colorful variation from red to yellow, at last to green, from the excitation spots to the end when excitation with 980 nm laser. However, the same nanoparticles in the capillary showed green color emission from start to end when shifting excitation light to 808 nm. In the case of $NaYbF_4:Er@NaYF_4:Yb(90\%)@NaNdF_4:Yb$, the quartz capillary demonstrated colorful luminescence variation from red to yellow and green upon excitation with 980 nm laser or 808 nm laser (Fig. 3f). Therefore, it is obvious that the doping amount of migrating ions in the middle layer makes a significant role on transferring energy from outmost layer to activator and modulator ions in the core layer, as in the case of Ho–Ce CR system. Meanwhile, decreasing the Yb doping ratio to 19% in core lattice inhibited the occurrence of the CR process between Yb and Er ions. Variation of migrating ions concentration in the middle layer did not influence luminescence colors no matter excitation at 980 or 808 nm (Supplementary Figs. 21 and 22). Both the nanoparticles $NaYF_4:Yb/Er$ (19/1%) @ $NaYF_4:Yb(10\%)@NaNdF_4:Yb$ and $NaYF_4:Yb/Er$ (19/1%)$@NaYF_4:Yb(90\%)@NaNdF_4:Yb$ in quartz capillary exhibited green color emission at 980 or 808 nm excitations.

In this Yb–Er CR system, Yb ions as an absorber of excitation photons participate in the CR process with activator Er ions. Thus, the energy of the excitation photon will directly influence the exciton populations of its $^4F_{9/2}$ states. In this work, it is easy to understand that the excitation power density decreases along with the increasing distance from the laser-emitted spot. To reveal this novel luminescence modulation in the capillary, we systematically studied the intensity ratio variations of red to green (R/G) light towards the excitation power. As Fig. 3g showed, R/G ratios of $NaYbF_4:Er@NaYF_4:Yb(10\%)@NaNdF_4:Yb$ negligibly changed upon exciting with different power of 808 nm laser, but exhibited a small extent variation from 2.7 to 5.2 when the power of 980 nm laser changed from 1.7 to 4.8 W. In contrast, the R/G ratios of $NaYbF_4:Er@NaYF_4:Yb(90\%)@NaNdF_4:Yb$ showed a different variation manner upon increasing laser power, that changed from 4.3 to 9.3 with 980 nm excitation, and from 1.8 to 3.5 with 808 nm excitation. Although the power increase seems to only induce a doubling of the R/G ratio in these cases, the luminescent colors exhibit a totally different variation because R/G ratios directly influence emission colors. As Supplementary Table 1 demonstrated, when Yb ions were doped with a ratio of 10% in the middle migration layer, the UCNPs exhibited green-colored emission both at 1.7 and 4.8 W of 808 nm excitation, but changed from yellowish color to red color when the power of 980 nm laser varies. In the case of 90% of Yb

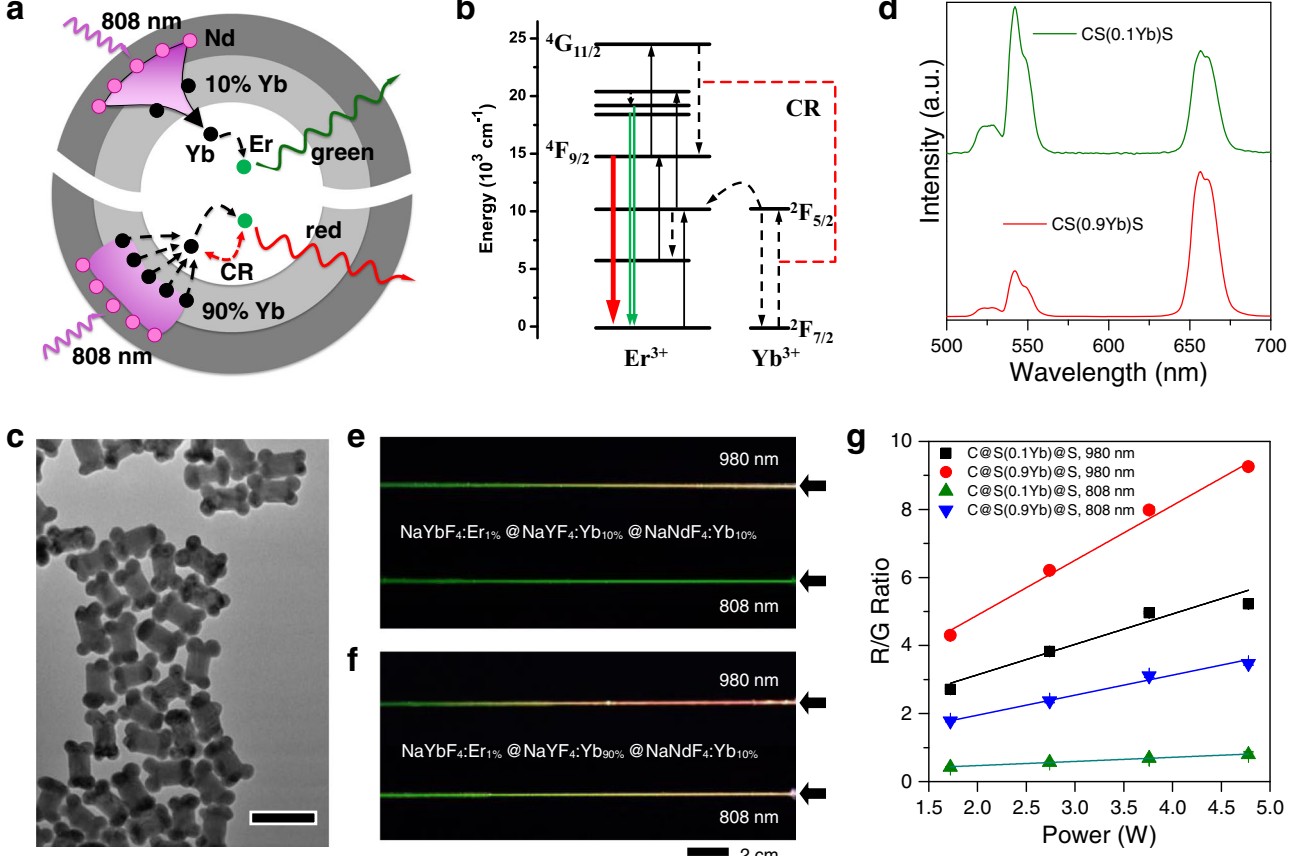

**Fig. 3 | Luminescence modulation of Yb–Er system. a** Different energy transfer process in the nanoparticles NaYbF$_4$:Er@NaYF$_4$:Yb@NaNdF$_4$:Yb when changing Yb doping ratio in the migrating layer. **b** Scheme for cross-relaxation process between Yb ions and Er ions. **c** TEM image of the nanoparticles NaYbF$_4$:Er@NaYF$_4$:Yb (10%) @ NaNdF$_4$:Yb. Scale bar is 100 nm. **d** Luminescent spectra of the nanoparticles under 808 nm excitation when Yb doping ratio changed from 10% to 90% in the migrating shell. **e** Luminescent images of the nanoparticles NaYbF$_4$:Er@NaYF$_4$:Yb(10%)@NaNdF$_4$:Yb in quartz capillary under 980 and 808 nm excitation. **f** Luminescent images of the nanoparticles NaYbF$_4$:Er@NaYF$_4$:Yb(90%) @NaNdF$_4$:Yb in capillary under 980 and 808 nm excitation. Scale bar is 2 cm. **g** Intensity ratios of red emission to green emission of the nanoparticles NaYbF$_4$:Er@NaYF$_4$:Yb($x$%)@NaNdF$_4$:Yb under different 980 and 808 nm excitation power. Data are presented as mean values ±SD ($n$ = 3). Source data in Fig. 3d and g are provided as a source data file.

doped in the middle layer, the emission of UCNPs changed from yellowish-colored emission to red when the power of 808 nm laser changed, but showed red-colored emission both at high laser power and low power of 980 nm excitation.

Similar to the luminescence changes in capillary, R/G intensity ratios of NaYF$_4$:Yb/Er (19/1%)@NaYF$_4$:Yb(10%)@NaNdF$_4$:Yb showed very small variations at all excitation conditions (Supplementary Fig. 23). This characteristic luminescence modulation further validates energy migration process plays a pivotal role on the CR efficiency. Moreover, because excitation photons were directly transited from Yb to Er ions in this system, excitation frequency showed negligible influence on R/G intensity ratios (Supplementary Fig. 24).

**Three-dimensional encryption application**

The ability of the nanocrystals to emit variable emissions on demand in response to excitation wavelength, pulse frequency, and power provides a convenient way of implementing multi-spatial and temporal encryption. As we know, a quick response (QR) code is a machine-readable two-dimensional code[34–36]. A vast number of geometric shapes can be defined within a QR code system, allowing direct storage of numerical data, such as name, address, phone number, location details, and so on. Fluorescent materials have been broadly explored to print QR codes for encryption[6,37]. Although it is invisible under natural light, actually it is easy to counterfeit once the information is decoded by scanning with a camera-enabled mobile device

having a QR code scanner application, due to its steady and single fluorescent color. Endowing the luminescent QR code with more variable fluorescence signals will make the QR code information cryptographic. Herein, as a proof-of-concept, a simple pattern with two coaxial squares was first fabricated to demonstrate its luminescence variation. In this pattern, Ho–Ce co-doped nanoparticles, termed as UCNP A, were located in the out square, and Yb–Er nanoparticles (UCNP B) in the inner square.

Figure 4a demonstrates the operation principle of the three-dimensional excitation-dependent information encryption. After exploiting the complementary responses between excitation power, wavelength, and frequency, the pattern demonstrated different colors when exerting diverse three-dimensional excitation parameters, ($x$, $y$, $z$). The detailed setting values of each excitation parameter were listed in Supplementary Table 2. As shown in Fig. 4b, the external and inner squares could show different or identical colors from red to green or yellow.

Furthermore, through coding the commercial or official secrets into the QR code at specific excitation parameters with a computerized algorithm, the information is encrypted and only can be decrypted at the certain conditions of ($x$, $y$, $z$). For instance, as a proof-of-concept demonstration, the word "UCNP" was first encrypted into a QR code pattern which was printed out by use of UCNP A, meanwhile, the word "JNU" also was encrypted into another QR code pattern which was printed out at the same position by UCNP B (Supplementary Fig. 25). As

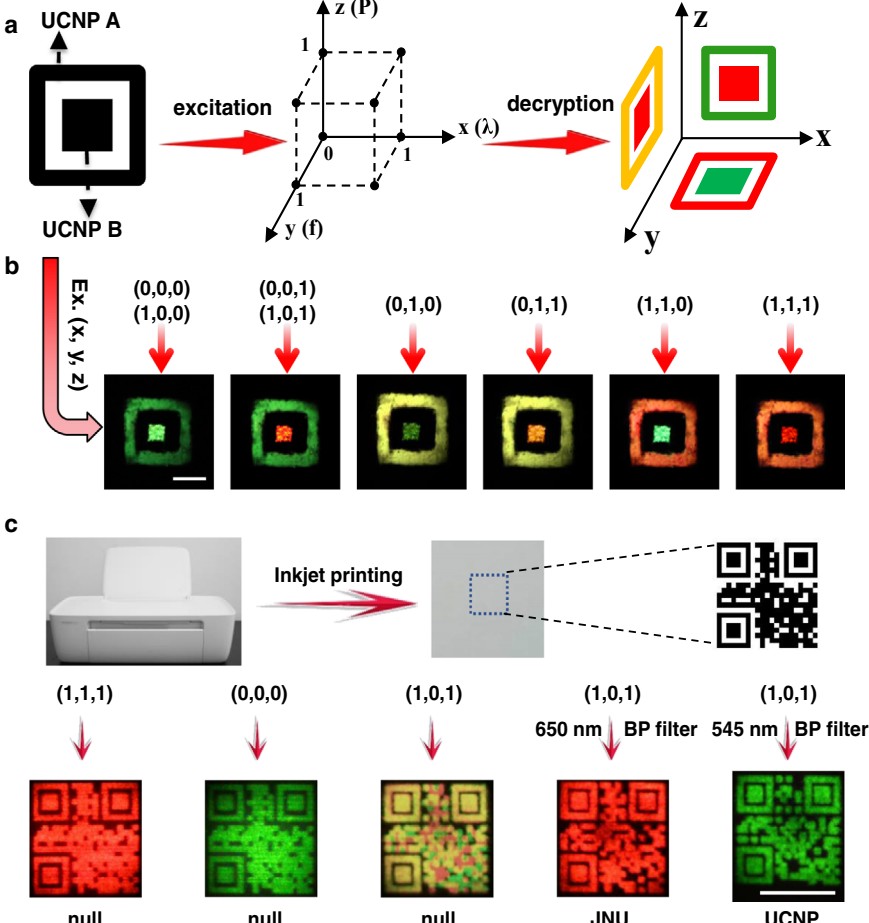

**Fig. 4 | Three-dimensional encrypting application by utilization of Ho–Ce nanoparticles and Yb–Er nanoparticles. a** The design principle of the three-dimensional encryption. UCNP A is the nanoparticle NaYF$_4$:Yb/Ho/Ce@NaYF$_4$:Yb@NaNdF$_4$:Yb. UCNP B is the nanoparticle NaYbF$_4$:Er@NaYF$_4$:Yb(90%) @NaNdF$_4$:Yb. **b** Luminescent pattern variations upon different three-dimensional excitation parameters. Scale bar is 1 cm. **c** Experimental demonstration of three-dimensional decryption of luminescent QR code by using these luminescence variable UCNPs. Scale bar is 1 cm.

shown in Fig. 4c, the QR code printed on paper was invisible under natural light, and exhibited different luminescent patterns when using diverse excitation parameters. The encoded information of "JNU" and "UCNP" only can be decrypted by scanning the QR code under a specific three-dimensional excitation manner (1, 0, 1) and passing through 545 nm band pass (BP) and 650 BP optical filter. Other luminescent patterns under the excitation conditions, such as (1, 1, 1), (0, 0, 0), and so on, cannot be decrypted to any information. Therefore, the encryption of secret information is greatly improved by the use of these luminescence variable nanoparticles.

## Discussion

In summary, lanthanide ions' advantages of abundant discrete energy states were successfully exploited to achieve variable luminescence on a simple structured UCNP. Two typical CR systems with different photon transition processes were investigated to verify the proposed concept that the CR process was sensitive to and easy to be modulated through the absorbed excitation energy. Excitation wavelength, power, and pulse frequency would influence energy absorption and migration pathway in the CR process, thus leading to a change in the population of excited states of lanthanide activator ions and dynamically adjusting their emission profiles. Although showing a drawback of relative low emission efficiency that most UCNPs may encounter, which may lead to the usage of relative high-powered lasers, this observation of multiple excitation-mediated CR efficiency gains a deep insight into luminescence modulations of lanthanide ions. By

employing these nanoparticles with variable luminescence to fabricate QR codes, the encoding information can be high-leveled encrypted, showing its promising potential in the frontier applications of information security.

## Data availability

The data sets generated during and/or analyzed during the current study are available from the corresponding author upon request. Source data are provided with this paper.

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

## Acknowledgements

The work was supported by the grants from National Natural Science Foundation of China (No. 22074028 and 21675038 to Q.M., Nos. 81971740 and 31671011 to Y.Z.), the Fundamental Research Funds for the Central Universities of China (No. 21622106 to Q.M.), and Innovative Research Team of High-Level Local Universities in Shanghai.

## Author contributions

Q.M. and Y.Z. conceived the project. X.F., S.F., and P.W. synthesized the upconversion nanoparticles. Q.L., J.Z., and J.L. designed and performed the encryption applications. Q.M., C.C., W.L., H.W., and Y.Z. analyzed the results, prepared the figures, and wrote the manuscript. All authors participated in the discussion.

## Competing interests

The authors declare no competing interests.
