## [Peer Review File · Nature Communications]

This manuscript has been previously reviewed at another journal that is not operating a transparent peer review scheme. This document only contains reviewer comments and rebuttal letters for versions considered at *Nature Communications*.

REVIEWER COMMENTS

Reviewer #3 (Remarks to the Author):

I thank the authors for their continued revisions and measurements.

Regarding Fig 3g, my larger point is that the authors try to highlight a substantial difference between the two wavelengths in their response to power, but in all cases the power increase leads to doubling of the R/G ratio - I don't see any difference at all. For example, this just doesn't seem correct to me:

"As Figure 3g showed, R/G ratios of NaYbF₄:Er @ NaYF₄:Yb(10%) @ NaNdF₄:Yb negligibly changed upon exciting with different power of 808 nm laser, but exhibited a small extent variation at 980 nm excitation. In contrast, the R/G ratios of NaYbF₄:Er @ NaYF₄:Yb(90%) @ NaNdF₄:Yb showed a markedly variation degrees both at 980 nm and 808 nm excitation when Yb concentrations increased to 90% in the middle layer, which demonstrated R/G ratios changed from 4.3 to 9.3 with 980 nm excitation, and from 1.8 to 3.5 with 808 nm excitation."

Regarding the demonstration, I do ask first that the authors be more forthcoming about the limitations of the measurement - 4-5 W of laser power is not something easily available in normal situations! I see what the authors are saying about the (x,y,z) parameters, and that is interesting, although as noted by other reviewers, not the first demonstration of these types of relationships, but a intriguing demonstration nonetheless.

Reviewer #3 (Remarks to the Author):

Comment 1. I thank the authors for their continued revisions and measurements.

Regarding Fig 3g, my larger point is that the authors try to highlight a substantial difference between the two wavelengths in their response to power, but in all cases the power increase leads to doubling of the R/G ratio - I don't see any difference at all. For example, this just doesn't seem correct to me:

"As Figure 3g showed, R/G ratios of NaYbF₄:Er @ NaYF₄:Yb(10%) @ NaNdF₄:Yb negligibly changed upon exciting with different power of 808 nm laser, but exhibited a small extent variation at 980 nm excitation. In contrast, the R/G ratios of NaYbF₄:Er @ NaYF₄:Yb(90%) @ NaNdF₄:Yb showed a markedly variation degrees both at 980 nm and 808 nm excitation when Yb concentrations increased to 90% in the middle layer, which demonstrated R/G ratios changed from 4.3 to 9.3 with 980 nm excitation, and from 1.8 to 3.5 with 808 nm excitation."

Response: Thanks for the reviewer's comments. Although the power increase seems to only induce doubling of the R/G ratio in these cases, luminescent color of the UCNPs demonstrates a totally different variation because R/G ratios directly influence emission colors. To clearly explain the different response to laser power, we have provided a table to summarize the luminescent color-variations of the Yb-Er based UCNPs at different excitation conditions.

As Table R-1 showed, when Yb ions doped with ratio of 10% in the middle migration layer, the UCNPs exhibited green-colored emission both at high laser power and low power of 808 nm excitation, but changed from yellowish color to red color when the power of 980 nm laser varied from 1.7 W to 4.8 W. In the case of 90% of Yb doped in the middle layer, the emission

of UCNPs varied from yellowish-colored emission to red when the power of 808 nm laser changed from 1.7 W to 4.8 W, but showed red-colored emission both at high laser power and low power of 980 nm excitation. To make it more clearly for readers, this paragraph has been revised as following.

"As Fig. 3g showed, R/G ratios of NaYbF₄:Er @ NaYF₄:Yb(10%) @ NaNdF₄:Yb negligibly changed upon exciting with different power of 808 nm laser, but exhibited a small extent variation from 2.7 to 5.2 when the power of 980 nm laser changed from 1.7 W to 4.8 W. In contrast, the R/G ratios of NaYbF₄:Er @ NaYF₄:Yb(90%) @ NaNdF₄:Yb showed a different variation manner upon increasing laser power, that changed from 4.3 to 9.3 with 980 nm excitation, and from 1.8 to 3.5 with 808 nm excitation. Although the power increase seems to only induce doubling of the R/G ratio in these cases, the luminescent colors exhibit a totally different variation because R/G ratios directly influence emission colors. As supplementary Table 1 demonstrated, when Yb ions doped with ratio of 10% in the middle migration layer, the UCNPs exhibited green-colored emission both at 1.7 W and 4.8 W of 808 nm excitation, but changed from yellowish color to red color when the power of 980 nm laser varied. In the case of 90% of Yb doped in the middle layer, the emission of UCNPs changed from yellowish-colored emission to red when the power of 808 nm laser changed, but showed red-colored emission both at high laser power and low power of 980 nm excitation."

Table R-1. Luminescent color comparisons of the Yb-Er based UCNPs at different excitation conditions.

	980 nm		808 nm	
	1.7 W	4.8 W	1.7 W	4.8 W
NaYbF ₄ :Er @ NaYF ₄ :Yb(10%) @ NaNdF ₄ :Yb				NaYbF ₄ :Er @ NaYF ₄ :Yb(90%) @ NaNdF ₄ :Yb				
Comment 2. Regarding the demonstration, I do ask first that the authors be more forthcoming about the limitations of the measurement - 4-5 W of laser power is not something easily available in normal situations! I see what the authors are saying about the (x,y,z) parameters, and that is interesting, although as noted by other reviewers, not the first demonstration of these types of relationships, but a intriguing demonstration nonetheless.

Response: Thanks for the reviewer's comments. As the referee mentioned, this work is an intriguing study of relationship between luminescence property and these (x,y,z) parameters. In this work, we propose that migrating ions in middle layer play a significant role in modulating CR efficiency in core nanocrystals when harvesting excitation energy at out layers, just liking an energy valve. Moreover, although a few works have investigated the relationship of luminescence properties with some of these excitation parameters, systematical and simultaneous study of their influence towards luminescence property of single activator ion is

rarely reported. This work provides an illuminating insight into mechanistic understanding and finely modulation of photon transitions in lanthanide-based upconversion.

For the concern of laser power, we believe it is not difficult to obtain a 4-5 W of laser power nowadays. A commercial 980-nm or 808-nm solid laser with power adjusting from 0 to 10 W only costs about 2 000 \$.

Reviewer #3 (Remarks to the Author):

Comment. I thank their authors for the continued efforts. I stand behind that the doubling effect does not demonstrate precise scientific control, but I'll agree that the particular colors vary as the authors say.

I also stand behind my statement about the challenges of high powered lasers in real use. Although the lasers are not particularly expensive, they are Class IV and require a large amount of care to be implemented safely and appropriately.

Response: Thanks for the reviewer's comments and reminding. More precise and scientific control of the energy migration process should be further investigated as the reviewer indicated, which is also the research objectives of our future works. In addition, the relative low emission efficiency may be the main drawback, that we have depicted in Summary section, leading to the usage of relative high-powered lasers. For the safety use of lasers, we have added the warning expressions in the Supplementary Information as following, "Cautions: Only trained personnel can operate lasers. Ocular and skin exposure to laser radiation may pose the risk of eye injury and skin burns. Enclose the laser setup in a box and wear laser safety goggles at 980 nm or 808 nm when operating lasers." Please refer to the last paragraph on Page 4 of the revised Supporting Information.